# The Development of an Experimental Framework to Explore the Generative Design Preference of a Machine Learning-Assisted Residential Site Plan Layout

**Pei Sun [1], Fengying Yan [1,\*], Qiwei He [1] and Hongjiang Liu [2]**

[1]  School of Architecture, Tianjin University, Tianjin 300072, China; sun_814224622@tju.edu.cn (P.S.); fslight@tju.edu.cn (Q.H.)

[2]  China Architecture Design & Research Group, Beijing 100044, China; hongjiang@tju.edu.cn

[\*]  Correspondence: fengying@tju.edu.cn; Tel.: +86-139-2030-9555

**Abstract:** Generative design based on machine learning has become an important area of application for artificial intelligence. Regarding the generative design process for residential site plan layouts (hereafter referred to as "RSPLs"), the lack of experimental demonstration begs the question: what are the design preferences of machine learning? In this case, all design elements of the target object need to be extracted as much as possible to conduct experimental studies to produce scientific experimental results. Based on this, the Pix2pix model was used as the test case for Chinese residential areas in this study. An experimental framework of "extract-translate-machine-learning-evaluate" is proposed, combining different machine and manual computations, as well as quantitative and qualitative evaluation techniques, to jointly determine which design elements and their characteristic representations are machine learning design preferences in the field of RSPL. The results show that machine learning can assist in optimizing the design of two particular RSPL elements to conform to residential site layout plans: plaza paving and landscaped green space. In addition, two other major elements, public facilities and spatial structures, were also found to exhibit more significant design preferences, with the largest percentage increase in the number of changes required after machine learning. Finally, the experimental framework established in this study compensates for the lack of consideration that all design elements of a residential area simultaneously utilize the same methodological framework. This can also assist planners in developing solutions that better meet the expectations of residents and can clarify the potential and advantageous directions for the application of machine learning-assisted RSPL.

**Keywords:** machine learning; generative design preference; planning design elements; Pix2pix model; residential site layout planning; experimental framework

## 1. Introduction

With the growth of computer science, machine learning-based generative design has become popular. This gives us new ways to learn about the generative design process for RSPLs. Generative design is performed using a computer that generates new design solutions in a given design space structure via random noise sampling. Machine learning, as a data-driven approach, is considered an effective method to apply to generative design [1,2]. The current generative design, which is built on machine learning, has met the need for devising a great number of design ideas. However, in the age of big data, most automatic design methods only look at quantitative goals and constraints and ignore qualitative design information, which is hard to describe mathematically [3,4].

Mining the generative design preferences of machine learning in the field of plan layout can help determine the design inspirations of machine learning in the field of RSPL design to explain the benefits of a machine learning-assisted plan layout. Generative design

preferences can guide planners in developing solutions that better meet residents' expectations. In recent years, investigation into the independent learning of generative preference design has begun regarding machine learning-assisted plan layouts. This investigation has two main paths for applying technical means and innovating research perspectives: (1) Machine learning is a technical tool used for solving research problems. For example, satellite images were used to identify land use changes [5], crime was assessed through street images [6], COVID-19 plan distribution states for urban security risk assessment were identified [7,8], and remote sensing images were used to detect forest carbon stocks to predict their carbon sink development [9]. (2) Machine learning provides innovative perspectives on extracting design elements to facilitate decisions. For instance, Silva et al. used convolutional neural networks and YOLO algorithms to identify sites and extract required elements (e.g., vegetation strips and buildings) to improve decision making for urban design development [10]. Moreover, Chinazzi et al. employed machine learning models to create a new method for generating scientific maps of knowledge, providing a scientific method for classifying urban planning and other fields [11]. Additionally, the creation of urban knowledge systems has been seen as an innovative result of the mutual representation of artificial intelligence techniques and the extraction of targets [12]. These earlier works have shown a strong link between machine learning and plan structure in recognizing, perceiving, evaluating, and predicting. Additionally, they showed that machine learning offers new ways to use technology to extract design elements to help plan layouts during autonomous learning exploration. This process allowed planners to determine the best practices for machine learning to assist with plan layout.

Each design element of an RSPL can be a generative design preference for machine learning in residential layout planning and can exhibit an application value. Residential design elements refer to each component of an RSPL, including design elements such as housing, roads, landscapes, and green spaces. These are indispensable and important components in the planning and design of residential spaces. Existing research of machine learning-assisted RSPLs only involves the study of individual design elements. For example, Xinyu Cong used CGAN to generate residential area layouts [13], Dai et al. used the Gray Wolf optimization algorithm model to improve the impact of community public space promotion from a child's perspective [14], and Elariane used a machine learning model to evaluate real estate website API data to determine the characteristics of long-term rental apartment homes [15]. Therefore, we attempt to apply machine learning to RSPLs through an experimental study of the totality of the design elements in residential planning, allowing machines to learn autonomously to determine their preferred designs of interest and their characteristic properties.

In the current artificial intelligence boom, generative adversarial networks have derived many new development-powered models such as CycleGAN [16], Pix2pix_HD [2], Pix2pix, etc. However, after combing through the literature regarding the strengths and weaknesses of each generative adversarial network model (as shown in Table 1), it was found that the Pix2pix model outperforms the others in the image transcription and classification tasks [17]. The Pix2pix model was proposed by Phillip Isola et al. in 2017 based on GANs, the earliest image recognition and generation applications. The most significant difference between the Pix2pix model and previous GAN-derived models is that Pix2pix optimizes the original input method to an imaging approach, enabling the image-from-to-image learning process [18]. Its discriminator Patch design can reduce the dimension of the input image significantly, reducing the number of parameters and increasing the operation speed. This study then generates a one-to-one site plan of the settlement to discover its design preference through labeled graphs. The Pix2pix model principle is to realize one-to-one image mapping. In addition, the external sites of residential areas have different shapes and scopes. In contrast, the Pix2pix model has no limitations regarding image scale and size, thus allowing for an increase in the scalability of the Pix2pix model. Therefore, the Pix2pix model was selected for the research in this paper. However, at the same time, the Pix2pix model has the disadvantage of generating fuzzy and conflicting

images. The existing research provides the following solutions: (1) increase the details of labeled maps and (2) improve the quality of the parameters.

**Table 1.** Comparison of the advantages and disadvantages of generative adversarial modeling.

| Model | Advantage | Disadvantage | Reference Sources |
|---|---|---|---|
| Pix2pixGAN | A generalized approach to image-to-image translation | Generates images with blurred, conflicting characteristics | Fu, B., et al. [19] Zhao, C. W., et al. [20] |
| CycleGAN | Solves the problem that the Pix2Pix model requires image pairing | Low quality of generated images | Zhu J Y, et al. [21] |
| Pix2pix_HD | Higher quality of generated images | Still needs pair data | Chen, J. S., et al. [22] |
| StarGAN | Realization of multi-domain style image transformation | The image's label is entered into the model so that the attributes can be modified | Shen, Y., et al. [23] Choi Y, et al. [24] |
| InfoGAN | The characteristics of the generated data are controlled by setting the implicit encoding of the input generator. | Training is unstable, and its performance is susceptible to the prior distribution and the number of noisy hidden variables selected. | Wan, P., et al. [25] Chen X, et al. [26] |
| LSGAN | Solves the problem of training instability | Lack of diversity in generated images | Mao X., et al. [27] |
| ProGAN | Generates high-resolution images | Very limited ability to control specific features of the generated image | Karras T., et al. [28] |
| SAGAN | Generated images more closely resemble the original image | Poor quality of images for generating local autocorrelation | Zhang H., et al. [29] |

Current research on applying the Pix2pix model has not been extended to other residential design elements. The application of the Pix2pix model was initiated at the beginning of Chaillou's implementation of the apartment plan design process, involving 'building plan contour', 'layout within the contour', and 'addition of furnishings', using multiple Pix2pix optimization models [30]. Pix2pix models were later optimized to evaluate automated building simulation applications [31]. For example, David Newton explored the challenge of generating layouts for Corbusier-style houses with a limited sample size. He expanded the scope of analysis by introducing noise and rotation to enhance the training effectiveness of GAN models [32] Yu et al. utilized traditional Chinese architectural datasets to generate and identify building facades [33]. Additionally, Mostafavi et al. employed machine learning to predict illumination and spatial daylight autonomy based on residential building spatial layouts [34]. However, previous studies demonstrate that the Pix2pix model has not been widely used in the design of RSPLs. While Gu D. et al. used the Pix2pix model to evaluate wind damage to residential building windows for protection against wind damage [35], their study focused solely on a single residential element. Few studies have extended the application of this model to other residential design elements and explored its diverse potential within the realm of RSPL.

Previous research has confirmed that machine learning can assist in generating and optimizing RSPLs. However, given a machine in a residential site scheme, it is unclear what the preferred design of machine learning in a residential site scheme is. This makes it hard to determine where the benefits of machine learning-assisted RSPL lie. Integrating the widespread use of large-scale data-assisted plan layouts and extracting the characteristics of the design elements of residential site schemes allows researchers to look into empirical methods to understand the potential for its use in "RSPL" from a machine learning point of view. Thus, this will contribute to the urban planning discipline. Given the above, this study presents an experimental framework for exploring Chinese residential areas. This stems from the diversity of residential types in Chinese residential areas, reflecting the universality of the research results. The proposed experimental framework was applied for experimental demonstration with the Pix2pix model as the chosen generative adversarial

network model. This provides a method for exploring design preferences in residential layout planning and extending the application of the Pix2pix model in RSPLs.

The rest of the paper is organized as follows: Section 2 presents the experimental framework for this study, including the residential area design element extraction process and the data processor for analysis using this framework. To verify the effectiveness of machine learning for residential layout planning, its learning results and limitations in this study, as well as insights for future work are evaluated and discussed in Section 3; Section 4 illustrates the analysis and the conclusions.

## 2. Materials and Methods

### 2.1. Study Area

The sources of the residential schemes for this study were CAD drawings of completed residential schemes collected from major Chinese design websites (shown in Appendix A Table A1). From these sources, we selected 300 design schemes for the experiment from residential areas in various provinces in China. The residential schemes we chose were all established settlements with a size ranging from 40 to 80 hectares and a predominantly rectilinear dwelling arrangement. The housing types were categorized based on their height: low-rise, multi-story, medium-high-rise, and high-rise. The plans of some of the residential schemes are illustrated in Figure 1. On average, the selected sample had 10.7 floors, a mid-level building density of 31.3%, an average floor area ratio of 1.9, and an average study area of 67.86 hectares (as shown in Table 2). We simplified the schemes and corrected them for code non-compliance and apparent errors.

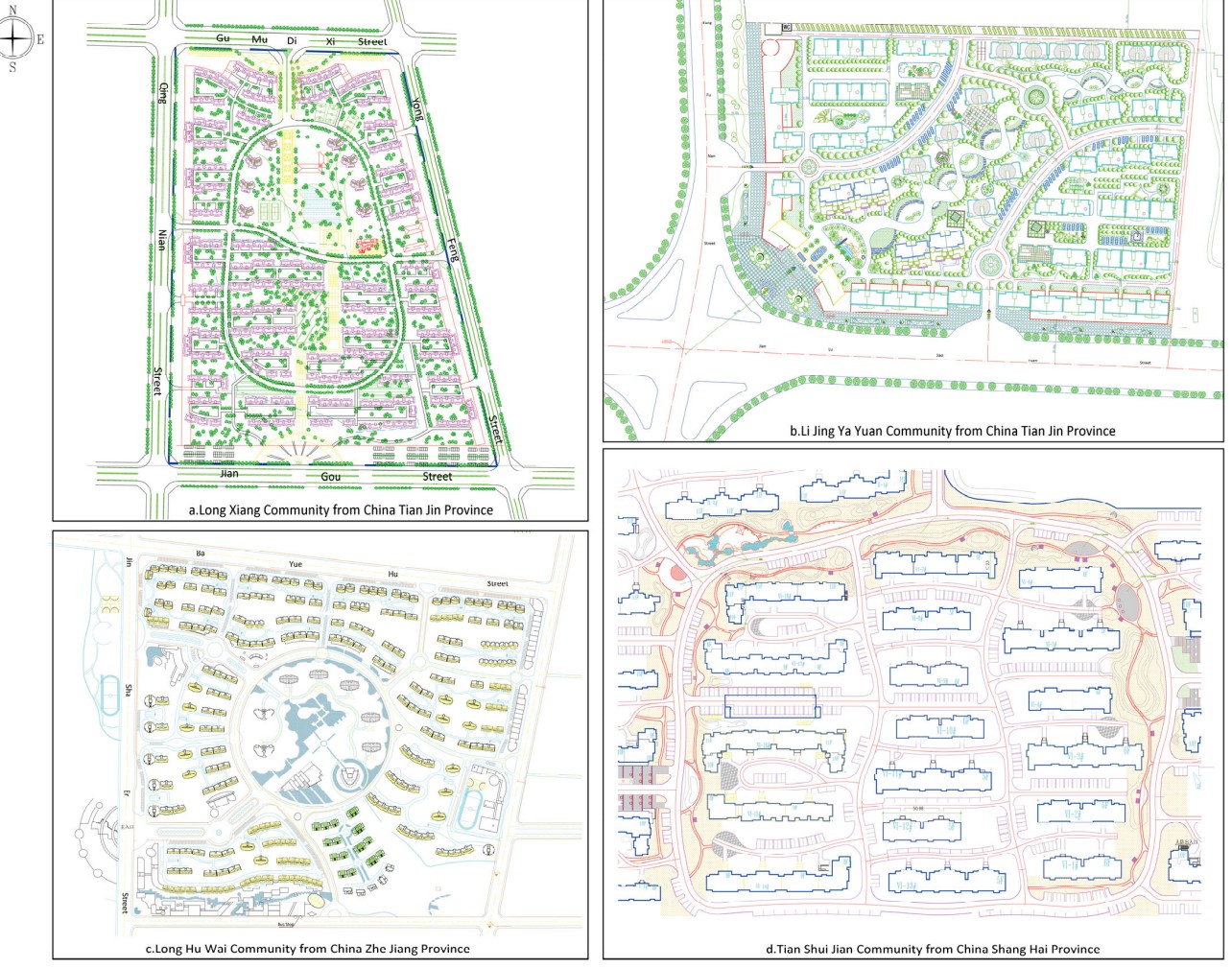

**Figure 1.** Scheme plans for residential areas.

**Table 2.** Summary of basic information in residential schemes.

| Basic Information Characteristics | Classification of the Basic Information Characteristics | Count |
|---|---|---|
| Floors | The highest number of floors | 32 F |
| | The lowest number of floors | 1 F |
| | Average floors | 10.7 F |
| Building density | Maximum building density | 39.4% |
| | Minimum building density | 20.1% |
| | Average building density | 31.3% |
| Plot ratio | Maximum floor area ratio | 4.4 |
| | Minimum floor area ratio | 0.7 |
| | Average plot ratio | 1.9 |
| Floor area | Maximum floor area | 87.3 ha |
| | Minimum floor area | 41.6 ha |
| | The average floor area | 67.86 ha |

### 2.2. Methodological Framework

To explore the "machine learning generative design preferences in RSPL.", a framework of "extraction-translation-machine learning-evaluation" was proposed (shown in Figure 2). The experimental framework is as follows: in the first step, design elements in China's Urban Residential Planning and Design Standard GB 50180-2018 [36] (hereafter referred to as "CURPADS") were summarized into five categories: housing, green space, supporting facilities, roads, and other elements. In the second step, we translated the scheme into an image recognized by the Pix2pix model using an RGB color block assignment of the image. The Pix2pix model was used in the third step to learn the residential area scheme, aiming to obtain an optimal parameter performance and a sample augmentation solution. Subsequently, the results of the generated solutions were evaluated through standard and design dimensions in the fourth step. The evaluation process represents the preferred design determination process.

### 2.3. Step 1: Extraction

Based on the current classification of "CURPADS", design elements of housing, green space, and other design elements (including square, water, inlet, and outlet), supporting facilities (commercial and other supporting facilities) and road elements were extracted as the design elements of RSPL that needed to be learned via machine learning for in-depth analysis in this study (as depicted in Figure 3). "CURPADS" has modified the requirements for the residential environment and supporting facilities. It incorporates housing, green space, and public space to enhance the quality of the residential environment, and it divides supporting facilities into different levels to align with the creation of residential areas of varying scales. In the latest residential design process, there is greater emphasis on improving the quality of the environment within residential areas while meeting mandatory design standards. Simultaneously, the quality of residential planning and design is ensured through a scientifically sound, green, and ecologically balanced spatial structure. "CURPADS" serves as the standard observed in Chinese residential planning and design. The residential design elements derived from it represent the accumulated practical experience of Chinese residential planning and hold significant importance.

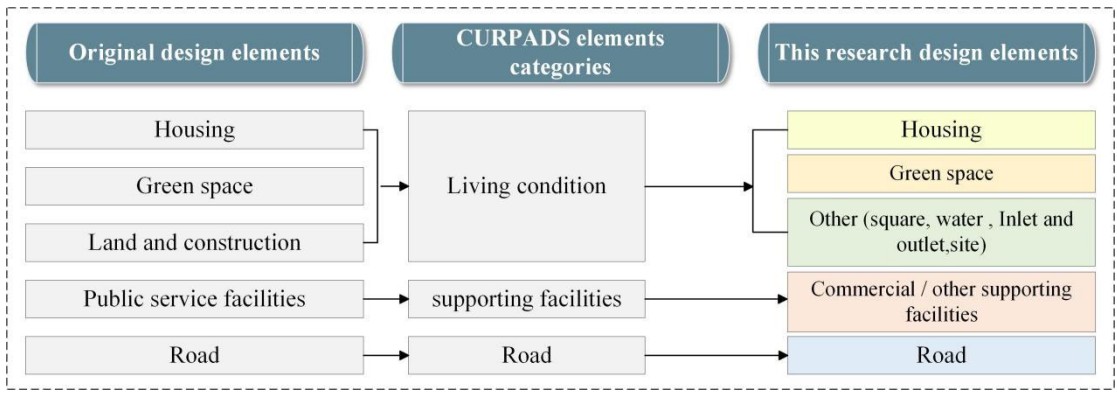

**Figure 2.** Experimental framework [36].

**Figure 3.** Classification of residential planning and design elements.

### 2.4. Step 2: Translation

Since the Pix2pix model performs image-to-image recognition, the residential design elements must be translated into residential images for machine learning. This is accomplished by translating the settlement design elements through assigning different RGB color values into images easily recognized and learned by the machine (as in Table 3). The final results of the labeled images are shown in a JEPG format with 512 pixels by 256 pixels and a resolution of 300 dpi.

**Table 3.** Different RGBs for different design elements.

| Extraction Elements | Function Type of Elements | RGB Value | |
|---|---|---|---|
| Housing | Villa (1–3 F) | R:80 G:120 B:80 | |
| | Low-rise (4–6 F) | R:255 G:0 B:255 | |
| | Mid-rise (7–11 F) | R:150 G:100 B:75 | |
| | Mid-rise (12–18 F) | R:180 G:0 B:255 | |
| | High-rise (over18 F) | R:255 G:150 B:150 | |
| Supporting facilities | Commercial supporting facilities | R:150 G:255 B:255 | |
| | Other supporting facilities | R:255 G:150 B:0 | |
| Road | External road | R:255 G:0 B:0 | |
| | Internal road | R:150 G:150 B:150 | |
| Green space | Greenery landscape | R:150 G:255 B:150 | |
| Other | Water | R:0 G:0 B:255 | |
| | Site | R:0 G:0 B:0 | |
| | Square | R:150 G:150 B:0 | |
| | Inlet and outlet | R:255 G:255 B:0 | |

### 2.5. Step 3: Machine Learning

#### 2.5.1. Pix2pix Model

The model used for this machine learning is Pix2pix, which operates with the underlying logic of a U-NET architecture [37] and consists of 16 layers of convolutional neural networks for the generator and a PatchGAN architecture [18], as well as five layers of convolutional neural networks for the discriminator (shown in Figure 4). The generator extracts the input image information containing various elements through a convolutional neural network. It conducts it through 16 different layers of neural networks, one layer at a time, to translate the image information into computer language before passing it to the next layer. Later, after receiving the training data forward propagated by the inputter through the deconvolution layer, the generated image is transmitted to the discriminator and bridged to the input image to determine the similarity of the generated image to the input image.

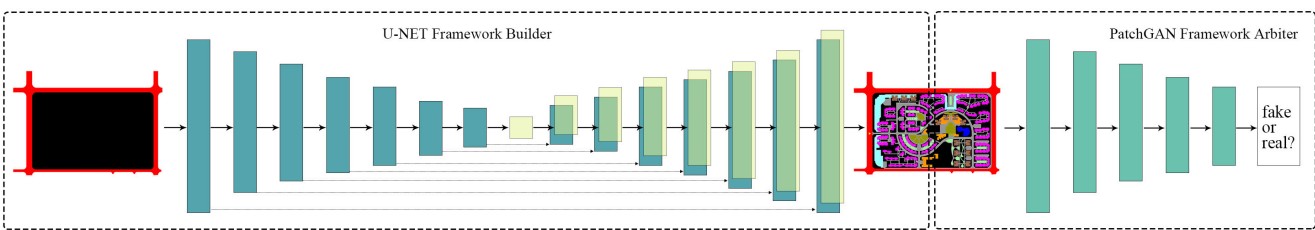

**Figure 4.** Model architecture diagram of Pix2pix.

#### 2.5.2. Learning Process

To maintain the optimal learning effect of machine learning during the experiment, the Pix2pix model needs to be optimized via multiple debugging. A machine learning process

under the computational mutual feedback system (shown in Figure 5) was proposed. This process comprises two parts: one is the tuning calculation, i.e., the parameter adjustment to determine its optimal parameters, and the other is the mutual data feed, i.e., the internal data augmentation to optimize its learning results.

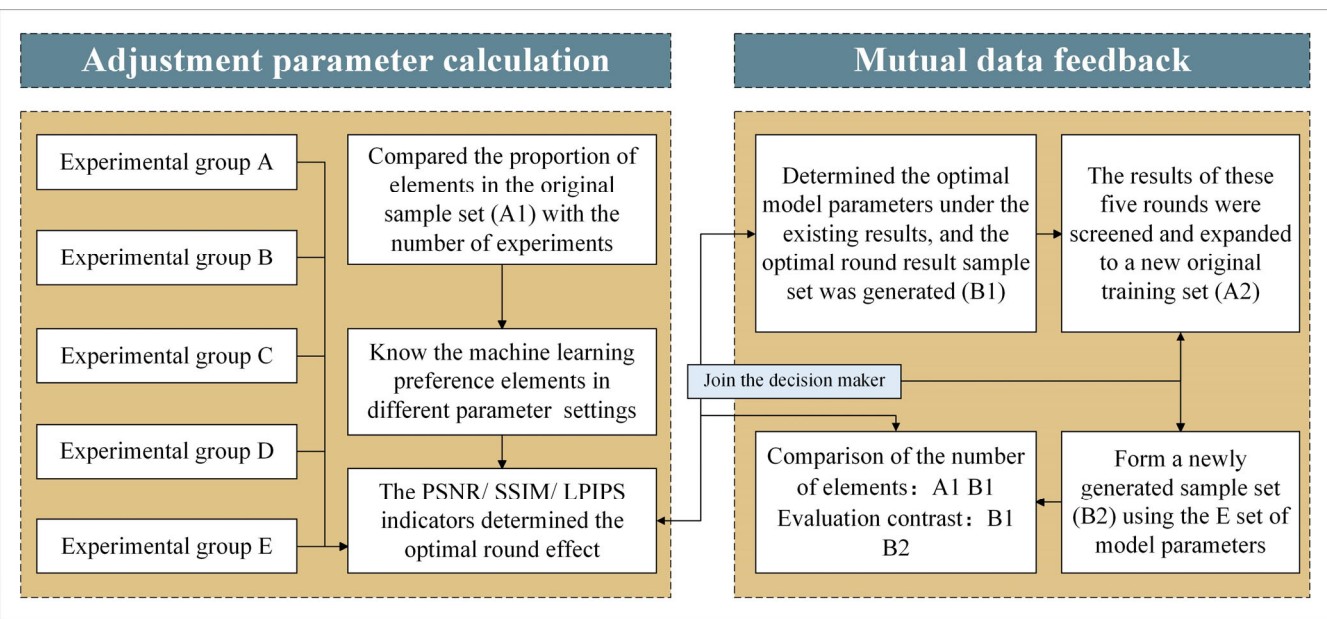

**Figure 5.** Flowchart of the calculated mutual feeding system.

- Parameter adjustment:

Parameters: loss function, hyperparameters, and metrics. The loss function referred to a function within the model, and the hyperparameters and metrics were used to tune the model and measure its performance, respectively. A total of five groups of tuning experiments, A, B, C, D, and E were carried out in this experiment.

Hyperparameters can influence the training and output performance of the model. Two main parameters were involved in this experiment: Epoch and Decay.

a. Epoch. The Pix2pix model learns all samples once during the learning process. A complete cycle is called one Epoch, through which the whole training process of the model is divided into several segments, and more iterations indicate a better learning effect. In this study, we selected epoch values of 100, 300, 500, and 700 for setting.

b. Decay. The decay degree represents the decay rate of the learning rate during the iterative process, and its purpose is to prevent overfitting. The optimal learning rate, which was immense initially and gradually decreased during the training process, could better approximate the optimal point. In the current work, we selected 50, 200, 250, 150, and 100 decay values for the setting.

Metrics were employed to evaluate the performance of different model algorithms. PSNR, SSIM, and LPIPS were chosen as the metrics for this experiment.

c. PSNR: Peak signal-to-noise ratio is a reference value of image quality that measures the difference between the maximum signal and background noise. It is the most common and widely used objective evaluation index for images and is usually defined by the sum mean square error (MSE) of the image. In detail, MSE is expressed as

$$MSE = \frac{1}{H \times W} \sum_{x=1}^{H} \sum_{y=1}^{W} (X(x,y) - Y(x,y))^2 \tag{1}$$

H aH and W represent, respectively, the length and width of the image, X denotes the original image, and Y indicates the generated image. $X(x,y), Y(x,y)$ represents the $(x,y)$ pixel value of the image X, Y in coordinates. PSNR is defined as [38]

$$\text{PSNR} = 10 \cdot \log_{10}\left(\frac{\text{MAX}_L^2}{\text{MSE}}\right) \tag{2}$$

where $\text{MAX}_L$ is the most probable maximum pixel value of the image. In the default red, green, and blue (RGB) images, this value equaled 255. MSE indicates the mean square error between the original and generated images. PSNR is measured in decibels (dB), and one of the objectives to this study is to generate image results with a high PSNR.

d. SSIM. The structural similarity index measure was used to compare the proximity of the original sample to the generated sample image with respect to brightness, contrast, and structure [39]. The SSIM algorithm was designed to consider the variation of structural information in the image in human perception [40]. The model also introduced perceptual phenomena and structural information related to perceptual variation. Structural information refers to the fact that pixels have internal dependencies on each other, especially spatially close pixel points [41]. These dependencies carry essential information about the visual perception of the target object, and therefore SSIM is more suitable than PSNR to evaluate the perceptual effects of images. Its definition is shown as

$$\text{SSIM}(x, y) = \frac{\left(2\mu_x\mu_y + \mathbb{C}_1\right)\left(2\sigma_{xy} + \mathbb{C}_2\right)}{\left(\mu_x^2 + \mu_y^2 + \mathbb{C}_1\right)\left(\sigma_x^2 + \sigma_y^2 + \mathbb{C}_2\right)} \tag{3}$$

where $\mu_x$ is the mean of x; $\mu_y$ indicates the mean of y; $\sigma_x$ and $\sigma_y$ are the variances of x and y; $\sigma_{xy}$ is the covariance of x and y; and $\mathbb{C}_1$ and $\mathbb{C}_2$ are constants. The proposed Pix2pix model aims to make the SSIM value as close to 1 as possible.

e. LPIPS. Learning Perceptual Image Block Similarity, also known as "loss of perception", was adopted to measure the difference between two images [42]. This metric learns the reverse mapping of the generated image to Ground Truth, forcing the generator to learn to reconstruct the reverse mapping of the real image from the fake image and prioritize perceptual similarity between them. LPIPS is more consistent with human perception than traditional methods (like L2/PSNR, SSIM, and FSIM). On the other hand, LPIPS can better reflect the perception advantage [43] of the images generated by GAN. A lower value of LPIPS indicates that the two images are more similar, and vice versa, the greater the difference. For a given neural network F, Figure 6 can represent how LPIPS is computed.

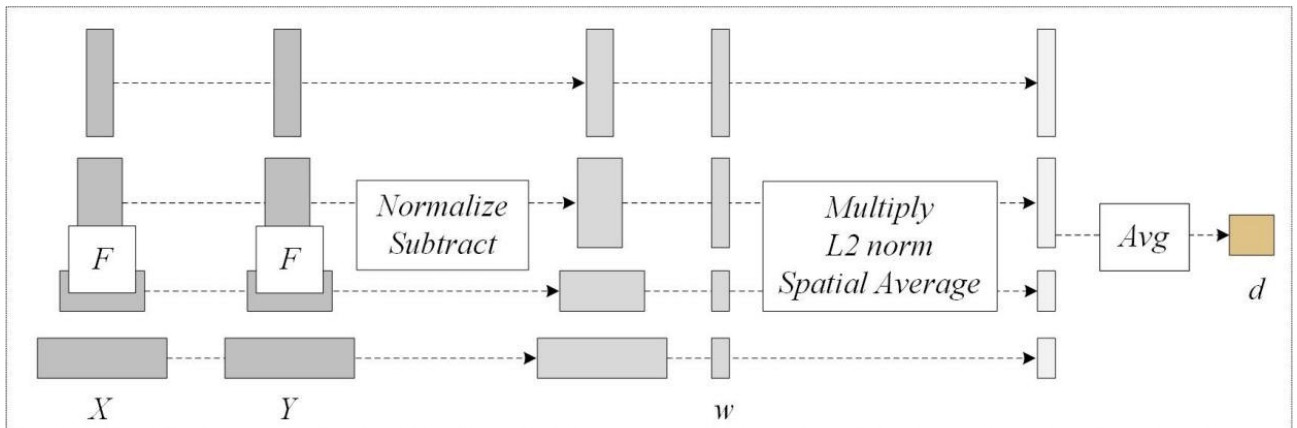

**Figure 6.** LPIPS operation process analysis.

The image was inputted to network F. Each convolutional layer was feature extracted and cell normalized in the channel dimension. For the L layer, the result would be written as $\hat{y}^l, \hat{y}_0^l \in R^{H_l \times W_l \times C_l}$. Meanwhile, each channel was scaled using the vector W and L2 distance was calculated. Finally, the perceptual distance result was obtained by averaging in the spatial dimension and summing in the channel dimension with the expression of

$$d(x, x_0) = \sum_\iota \frac{1}{H_\iota W_\iota} \sum_{\mathcal{H},\mathcal{W}} \| \mathcal{W}_\iota \odot (\hat{y}^\iota_{\mathcal{H}\mathcal{W}} - \hat{y}^\iota_{0\mathcal{H}\mathcal{W}}) \|_2^2 \tag{4}$$

- Data Enhancement:

Data augmentation refers to making a limited amount of data produce more value without substantially increasing the data [44]. In this work, the solution with a better effect on the generation side of experimental groups A, B, C, D, and E was added to the original sample to achieve sample augmentation. Combining the original sample and the generated sample increases the diversity of the data set, improves the generalization ability and robustness of the training model [45], and thus enhances the value of the existing data. For the screening of sample set augmentation, the following process was mainly adopted: (1) an overall judgment was made about whether the scheme was complete and whether each design element in each scheme was easily distinguishable, etc., (2) a judgment of each design element was performed (Figure 7). If a design element could not be judged, other design elements would be combined to make a comprehensive judgment. If the generated images conformed to the judgment process, they would be mixed into the original sample set. Otherwise, the solution would be filtered and discarded. Finally, we selected 285 images from the 1500 generated results (experimental groups A, B, C, D, and E) and blended them into the original sample set for data enhancement.

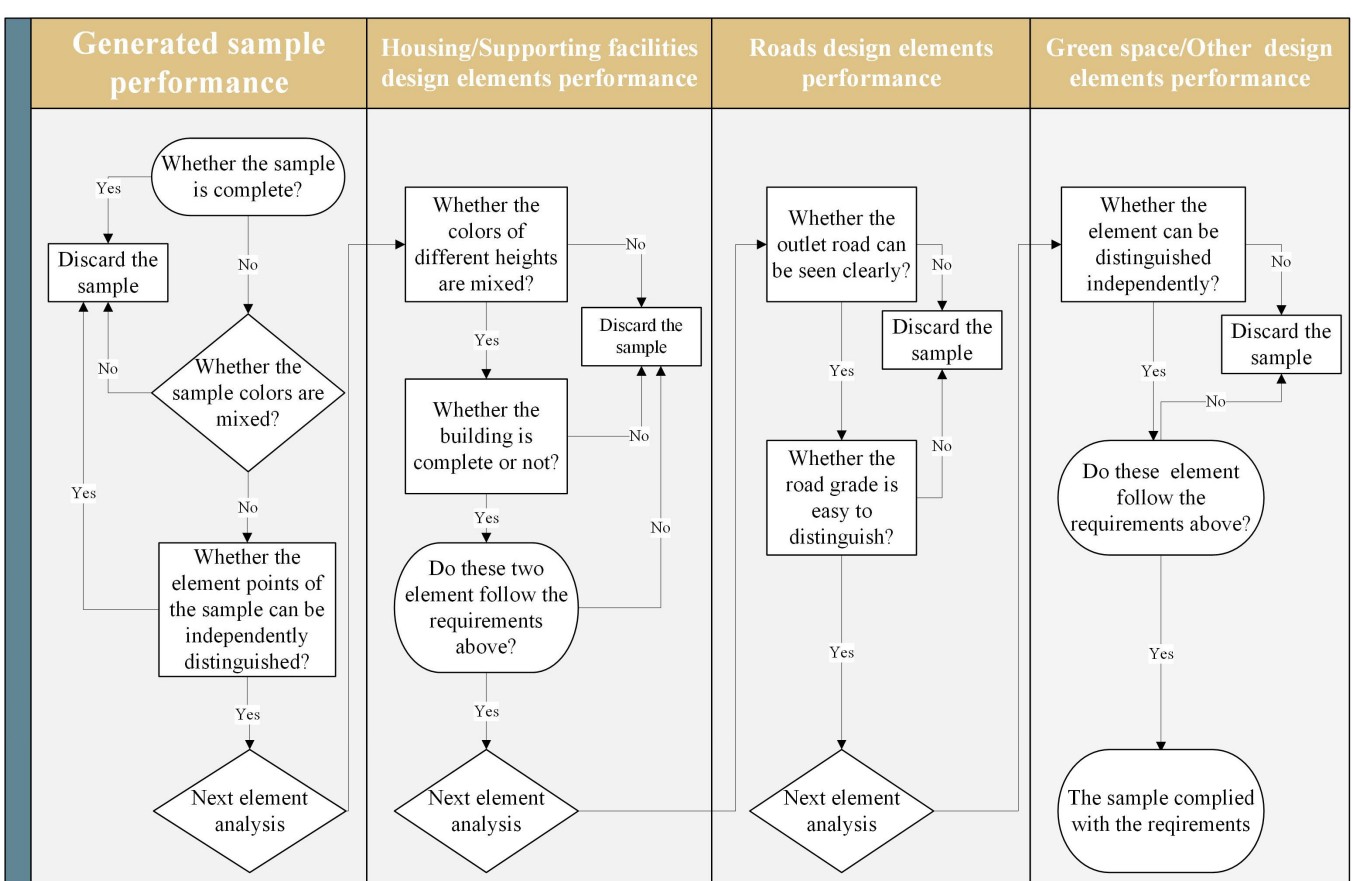

**Figure 7.** Sample set expansion and screening process.

### 2.6. Evaluation

Since machines cannot judge the quality of image design, each result in machine learning requires metrics for evaluation. According to Recio et al., it was found that in emotion, the high arousal effect performance of positive words leads to a faster visual perceptual response [46] and more easily obtained merits of the target object at the visual level. In addition, the positive evaluation words mentioned in the book "Designing Cities:

Basics-Principles-and Practice" were the first to utilize this evaluation in urban design [47] and were able to precisely identify design solutions. Hence, based on this reference, an evaluation scale for a design dimension was proposed in this study. The evaluation of this study was divided into two dimensions: the design dimension and the standard dimension. The design dimension proposes diversity, simplicity, relative property, and totality to evaluate the square paving, green landscape space, commercial facilities, and other public facilities (as shown in Table 4). For the standard dimension, three aspects of plot ratio and building density, as well as the proportion of square paving, green landscape space, commercial facilities, and other public facility activity-occupied land, were selected to evaluate the results.

**Table 4.** Classification of the evaluation dimensions.

| Evaluative Dimension | Evaluate Elements | Square Paving | Landscape Green Space | Commercial Facilities | Other Public Facilities |
|---|---|---|---|---|---|
| Design dimension | Diversity | ①Structured; ②Detailed; ③Various; | | | |
| | Simplicity | ①Well-balanced; ②Self-existed; ③Concise; | | | |
| | Relative property | ①Sequential; ②Heterogeneous; | | | |
| | Totality | ①Compact; ②Unified; ③Balanced; ④Uniform; | | | |
| Standard dimension | Plot ratio | Density of the building | The proportion of paved plazas/landscaped green areas/commercial facilities/other public facility activity sites | | |

## 3. Results and Discussion

### 3.1. Optimal Parameter Determination

We compared the experimental results in the five groups of parameters by selecting one of the residential area schemes (as in Figure 8). It was found that the PSNR and SSIM index scores reached the highest level in experimental group E, while the LPIPS index showed the lowest also in experimental group E. Subsequently, the generated results from the five experimental groups were compared using the mean opinion scoring method, which is a subjective image quality assessment index that rates the visual perceptual quality of the generated images on a scale from 1 (worst quality) to 5 (best quality). The final score is calculated as the arithmetic mean of the scores provided by all the raters. In this case, the highest mean score for Group E was 3.7, based on the ratings from 30 raters (see Appendix A Table A2). Based on these results, Group E is considered to have performed the best. Additionally, this group had the highest number of iterations, and the degree of decay was maintained in a gradually decreasing state. Consequently, these parameters will be used in the subsequent model training for scheme learning.

### 3.2. Generative Preference Design Element Determination

Once the parameters for extracting the generative design preferences of machine learning in the design elements of "RSPL" were determined, a visual method was employed to quantitatively compare the number of element changes between the original sample A1 and the generated sample B1 (as shown in Table 5). Since not all residential schemes contain all design elements, it is necessary to perform a classification count before tallying the number of element changes. The statistics are as follows: water and supporting infrastructure are classified to initially count the number of original sample sets with or without such elements. Subsequently, the number of schemes with or without these elements was counted through machine learning to discern the differences.

For example, in the original sample A1, the percentage of residential schemes with other supporting facilities was 39.7%. However, in sample B1, which was generated after machine learning, the percentage of residential schemes with commercial supporting facilities increased to 62.1%. Conversely, the proportion of residential schemes without other facilities in the original sample A1 accounted for 60.3%, while in sample B1 generated by machine learning, the proportion of residential schemes without commercial facilities

decreased to 39.7%. This indicates that the number of residential schemes with other supporting facilities increased after machine learning.

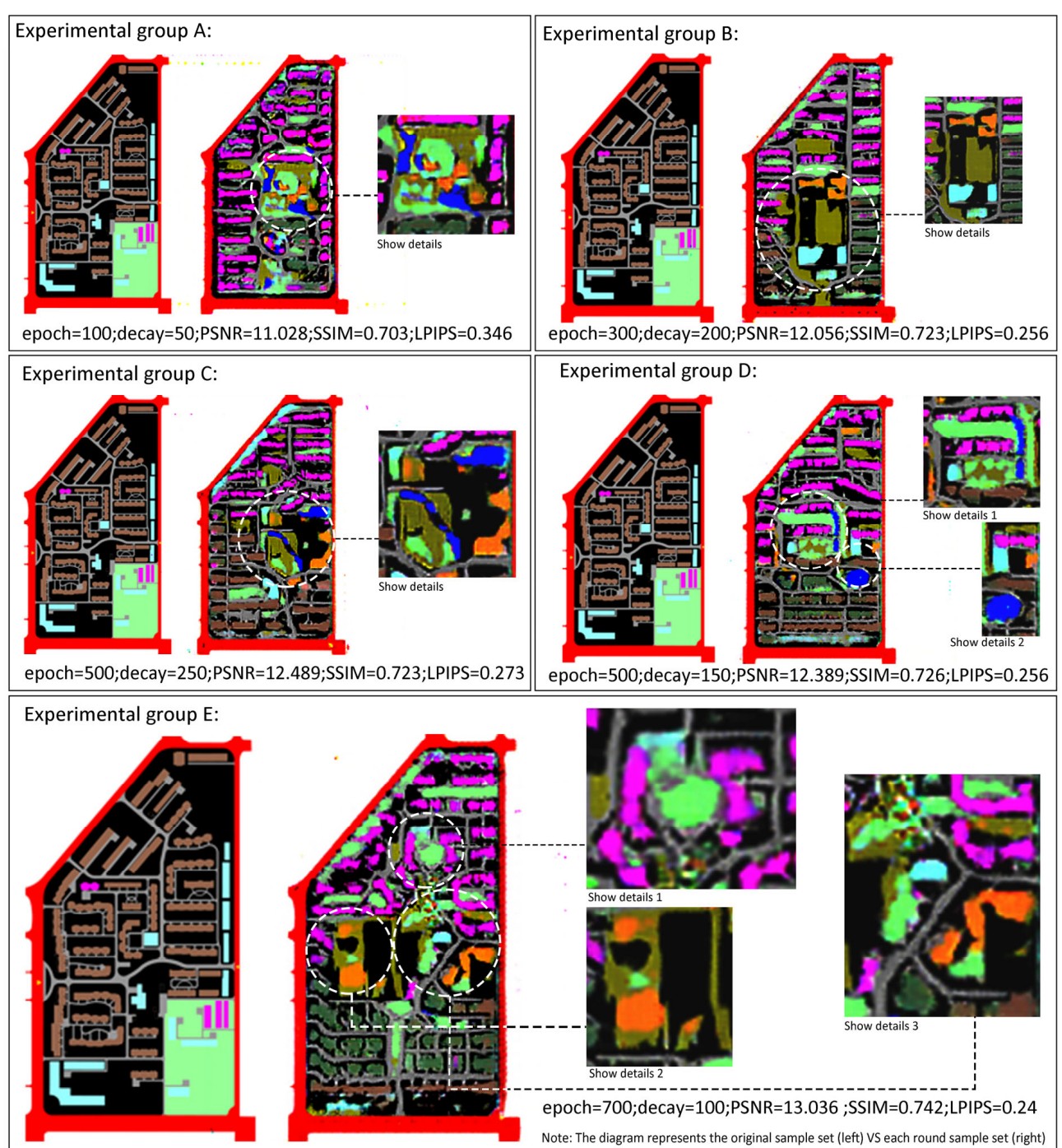

**Figure 8.** Partial table for generating the sample set.

Since spatial structure, landscaped green space, and road design elements are present in each sample, statistical classification is necessary for these three design elements based on their types. In this study, both elements were categorized into three groups: concentrated, dispersed, and centralized-dispersed, using a visual method. Statistics were conducted based on the difference in the number of types before and after their generation. For instance, in the original sample A1, 52.7% of residential schemes in green landscape regions were decentralized. After machine learning, 55.1% of residential schemes with decentralized landscape green areas were generated in sample B1. The proportion of

residential schemes with concentrated landscape green space increased from 8.5% to 11.6%. However, the proportion of residential schemes with scattered landscape green space decreased from 38.3% to 33.3%. In the original sample A1, 29.7% of residential schemes featured axial roads, while 41.8% featured axial roads in machine-learned sample B1. Furthermore, the residential schemes in sample B1 generated by machine learning reached 41.8%. In contrast to the residential schemes with an axial-ring road (17.3% to 0.7%), the proportion of residential schemes with a ring road increased from 53.0% to 57.5%. According to the above methodologies, only public facilities and spatial structure differed above 10% in the number of changes in all categories.

**Table 5.** Comparison of element classification and quantitative statistical results.

| Extraction Elements | | Classification of Elements | The Proportion of Elements in the Original Sample A1 | The Proportion of Elements in the Generated Sample B1 |
|---|---|---|---|---|
| Water | | Yes | 60.6% | 54.4% |
| | | No | 39.4% | 45.6% |
| Supporting facilities | Commercial supporting facilities | Yes | 60.3% | 78.6% |
| | | No | 39.7% | 21.4% |
| | Other supporting facilities | Yes | 39.7% | 62.1% |
| | | No | 60.3% | 37.9% |
| Road network structure | | Axis | 29.7% | 41.8% |
| | | Ring | 53.0% | 57.5% |
| | | Axis-ring line | 17.3% | 0.7% |
| Space structure | | dispersed | 32.0% | 58.2% |
| | | concentrated | 28.0% | 15.4% |
| | | centralized-dispersed | 40.0% | 22.8% |
| Landscape greening structure | | dispersed | 52.7% | 55.1% |
| | | concentrated | 8.5% | 11.6% |
| | | centralized-dispersed | 38.8% | 33.3% |

Based on the methodological statistics mentioned above, it is evident that machine learning exhibits a preference for designing two major elements: other public facilities and spatial structure. Delving deeper into the reasons for their prominence, our concept of other public facilities in this study is defined as independent and large-area facilities such as kindergartens, elementary schools, and cultural activity centers. These facilities offer certain advantages in the image translation process when compared to other public facilities: (1) There are no additional elements surrounding them that could cause interference. In fact, our original sample set indicates that most elements of other public facilities exist in isolated corners and do not blend with other elements, minimizing interference with machine learning; (2) Due to their larger size, other public facilities are also represented by RGB pixel values in the original sample set; (3) These facilities tend to exhibit better contrast, resulting in improved machine learning results due to more pronounced shaping. This study assessed spatial structure based on the combination of square paving and green landscape elements. Furthermore, these two elements exhibited various characteristics, such as fragmented connectivity, scattered distribution, and a substantial plan area, during our labeling process. A combination of square paving and green landscape was utilized to evaluate spatial structures. Machine learning for spatial structure design elements tends to generate decentralized spatial structures. The learning effect is more favorable because these two elements exhibit diverse characteristics, including fragment connectivity, scattered distribution, and a large plan area. While Ma et al. concluded that uniformly

distributed spatial facility service components were crucial in shared rental housing [48], a thorough analysis of the generated sample set by machine learning yielded a programmatic surface effect that prefers a balanced layout for each design element. This confirms that the results align with the flat characteristics of an RSPL [49].

*3.3. The Design Dimension Determines Preferred Generative Design Features*

Among the 12 display schemes selected (as in Figure 9), we chose two for successive comparative evaluations of the machine learning process. The original sample for Scheme 52 featured a core spatial structure comprising water and a square, with supporting facilities distributed to the right and left of the entrance. In contrast, the generated result sample 52-B1 exhibited significantly higher building density. Furthermore, it used other supporting facilities and water as the core spatial design elements within the residential area. Some commercial facilities were added to the south side to complement the design along the residential interface, but the design of the square landscape green space needed to be incorporated. On the other hand, the generated sample 52-B2 featured a core residential space composed of water, square landscaping, and other supporting facilities. It positioned commercial facilities and a larger other supporting facility on the side of the main road. The original sample 92 had a simpler design, with only water and small squares as the core space, along with some small supporting facilities distributed along the residential area. In sample 92-B1, the core residential space consisted of a large square and green area, which required more control over its scale due to its substantial size. The core residential space in sample 92-B2 was formed by other supporting facilities and a square landscape. While the inner ring road from the original sample was retained, a portion of the open square was designed by extending it along the left side of the main road towards the exterior. Upon comparing the two solutions above, it became evident that the generated sample B2 placed greater emphasis on shaping spatial structure and green landscape space compared to B1. Additionally, the overall solution was more mature, encompassing all the elements of a residential planning study. Its spatial structure and green landscape space were shaped with greater flexibility and diversity than the original sample, featuring better scale control and a more complete form.

After comparing and evaluating the generated results from parameter set E (B1) with the generated results (B2), which were obtained by mixing the training of the newly generated sample set of 285 solutions, it was evident that, from an overall perspective of diversity and contrasting learning, the design dimension was more effective in generating samples B1 and B2. This suggests that machine learning can generate innovative solutions and provide design ideas, aligning with the concept that machine learning can generate innovative building graphics and section designs through 3D models, as previously confirmed by other studies [50]. The results also verified that positive terms used for evaluating the solutions generated at the diversity level were primarily "structured" and "formally diverse". "Structured" implies that machine-learned solutions produced monocentric or polycentric spatial structures, while "diverse" signifies the diverse spatial structures formed by combining amenities, squares, and green spaces. It was noteworthy that the positive word "diverse" appeared more frequently in the sample B2 generation than in sample B1, suggesting that the performance of the data-enhanced design solutions was more inspiring to designers. At the "relative property" level, the positive words for the generated schemes were mainly "heterogeneous". Interestingly, some of the solutions that exhibited "sequential" positive words in generation sample B1 transformed into "heterogeneous" in generation sample B2 (as shown in Figure 10). This indicates that after data enhancement, machine learning for scheme results displayed more design flexibility and showcased the innovative potential of square pavement and landscaped green space to conform to the building layout. For example, Schemes 212 and 109 demonstrated the design flexibility of paving and landscaping in response to the building layout.

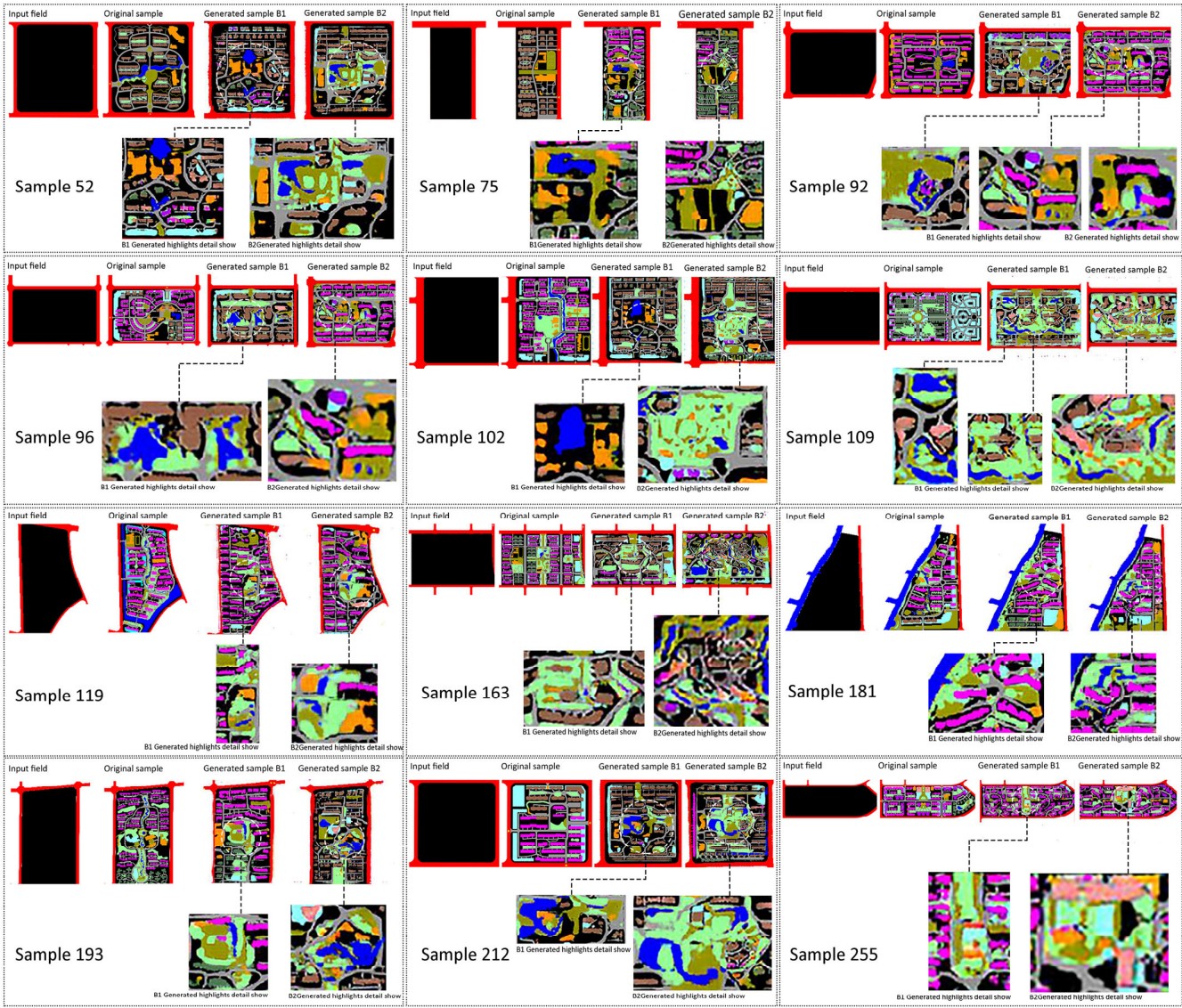

**Figure 9.** Partial display table for generating the sample set.

The results of this study primarily emphasize the autonomous exploration of RSPL in generative design preferences. In contrast, earlier studies focused on analyzing how machine learning can assist in optimizing and reconfiguring the spatial structure of planned designs [51]. This novel approach to applying machine learning in plan layouts allows for a more robust exploration of the potential of machine learning-assisted applications in RSPLs.

### 3.4. Standard Dimension Determines Generative Preferred Design Features

By analyzing the 12 selected display solutions and considering specification indicators such as plot ratio, building density, and active land use proportion, it became evident that the building density in the original sample set A1 and the generated sample sets B1 and B2 is identical and complies with the relevant design specifications (Figure 11a). However, the fluctuation range of the building density of the generated sample B2 is smaller than that of both the original and generated sample B1 (ranging from 25% to 39%). In contrast, the building density in the original sample A1 exhibits a fluctuation range between 20% and 39%. Although this plot ratio is lower than the original sample set, its fluctuation range is also smaller, maintained between 1.1–and 3 (as shown in Figure 11b). In contrast, the plan area ratio of the original sample A1 and the generated sample B1 has a fluctuation

range exceeding 3. This phenomenon is presumably linked to the selection of more mid-rise building height solutions, indicating that the machine learning-generated solutions predominantly feature mid-rise building heights. Nevertheless, it is also confirmed that the performance of the machine-generated residential area scheme becomes more stable after data enhancement.

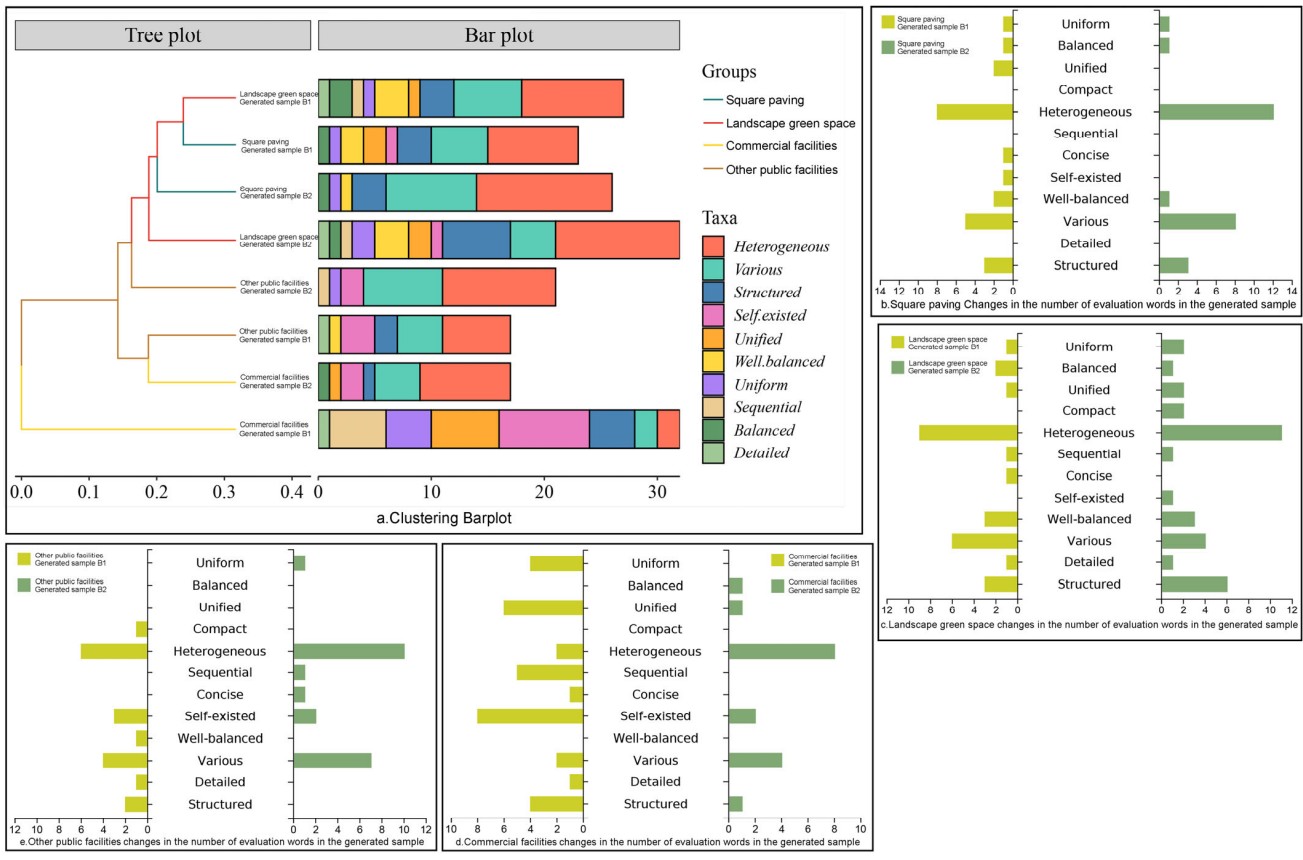

**Figure 10.** Evaluation results via graphs in the design dimension (Figure 9).

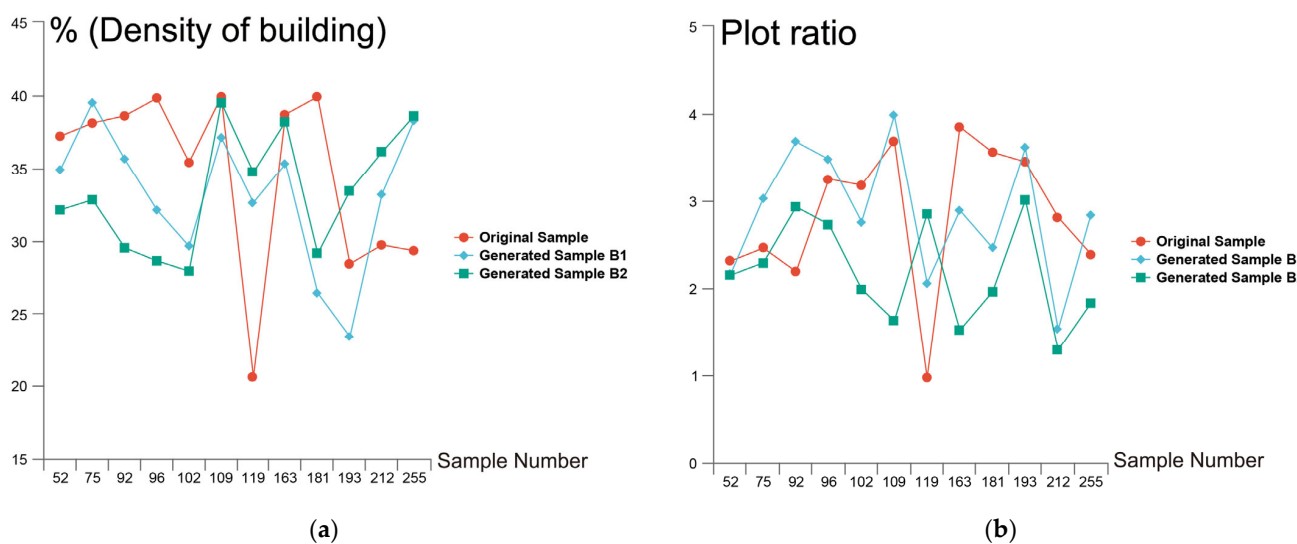

**Figure 11.** (**a**) Graphs of building density generated for some of the display samples in Figure 9; (**b**) Graphs of a plot ratio generated for some of the display samples in Figure 9.

Regarding the machine learning effect on the land of each activity, the generated square pavement and green landscape space became more or less concurrent compared to

the original sample. In contrast, the commercial and other supporting facilities appeared smaller or converged than in the original sample (Figure 8). However, according to the statistics presented in Table 2, the commercial facilities and other supporting facilities in the generated sample B1 increased compared to the original sample. It is important to note that these statistics pertain to the proportion of occupied land, demonstrating that the scale of commercial facilities and other supporting facilities in the generated samples B1 and B2 is smaller than that in the original sample. This aligns with the previously explained machine learning-generated design scheme, which pursues a more balanced layout effect. When comparing, it becomes apparent that a significant portion of the square paving and landscaped green space in generated sample B2 converged to a greater or lesser extent when compared to generated sample B1. Commercial facilities tended to be fewer or more condensed, while other supporting facilities mostly remained unchanged (Figure 12). It appears that the machine learning effect may have been more successful in replicating the commercial facilities lined up along the street in the machine learning scheme, as most of the generated schemes did not exhibit this particular performance characteristic. This could be attributed to the fact that most of the commercial facilities in this study were commercial facilities lined up along the street. However, the results of our generation sample B2 were influenced by a mixture of the five groups A, B, C, D, and E with better experimental results, which were then re-generated. Consequently, the likelihood that the machine needed to learn about commercial facilities increased. This might explain why the generated sample B2 had fewer converging commercial facilities compared to the generated sample B1.

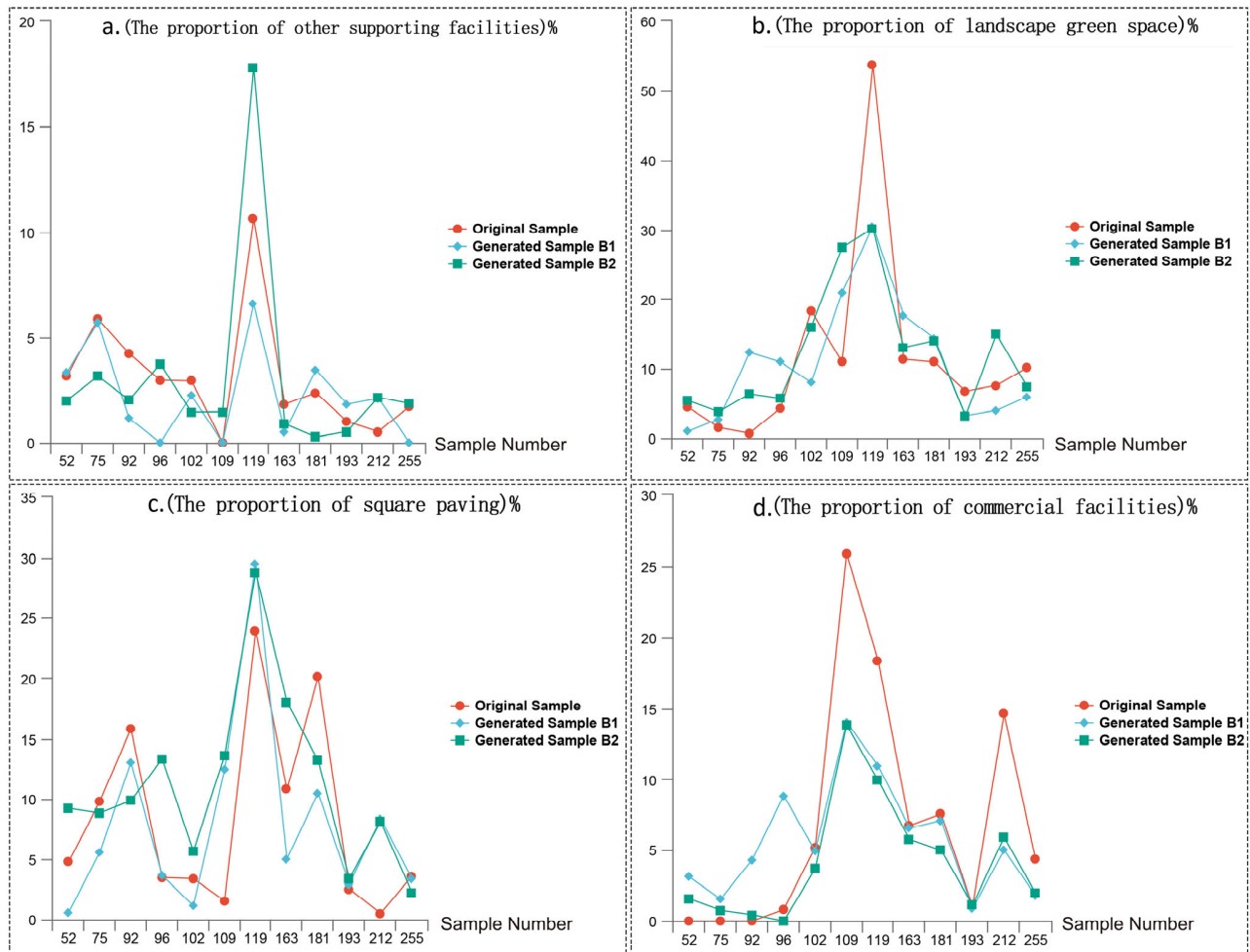

**Figure 12.** Each element of the active land proportion separately shown in part in Figure 9.

As mentioned earlier, following machine learning, residential design elements exhibit varying effects, with other public facilities and spatial structures proving to be the most influential and displaying diverse characteristics. Therefore, it is crucial to explore how to fully harness the potential and direction of applying design elements related to other public facilities and spatial structures. "CURPADS" focuses more on creating living circles in residential areas. It advocates the division of amenities into internal communities and settlements [52,53] and attaches more attention to public facilities to guide and define the division of residential communities. This study confirmed that the amenities in the generated scheme fulfill the role of guiding the spatial structure of the settlement that machine learning can achieve. Consequently, the machine learning preference design was considered to optimize the creation of living circles in residential areas. In studies concerning the configuration and layout of public facilities and green landscape spaces in urban planning, Khodaparasti et al. introduced an "integrated location-allocation" model to optimize the equity and efficiency of medical service facility locations [54]. Wang et al. used machine learning algorithms and POI data to select the location of elderly facilities in Wuhan [55]. The green space network of Lijiang City was constructed by Ren et al. using satellite images to design green space as a multimodal space of points, linear bars, and irregular shapes [56]. Also, the studies above can be used to analyze the design preferences and characteristics of residential areas found in this study. This allows for the studies above of public facilities, landscape green space layout, and different types of spaces to help create urban public facilities and landscape green spaces. As a result, their application potential would be expanded while contributing to the development of the urban planning discipline. In the future, we can explore additional applications of the Pix2pix model within RSPL as a specific application area, thus uncovering further value-added possibilities for the Pix2pix model.

Finally, this paper not only explores generative design preferences at the plan image design level but also considers how to enable the machine to discover generative design preferences for spatial design from the spatial planning level, which is a topic for future research experiments. Ideally, in the future, we will continue to optimize the performance of the Pix2pix model to enhance the stability of its training, leading to improved image resolution in generated designs. This will increase the generalizability of the Pix2pix model and expand its application value in the field of RSPL on a larger scale. In addition, we attempted to optimize the algorithm for LPIPS metrics to derive a machine learning visual perception evaluation metric that aligns more closely with human design thinking. The goal is to use this as a classification basis to score generated balanced layout surfaces, conduct an in-depth classification study of their balanced layout characteristics, and explore the applicability of balanced layout effects in various urban design schemes.

## 4. Conclusions

In this study, we conducted an experimental exploration of machine learning generative design preferences in RSPL using the Pix2pix model. The aim was to uncover machine learning's generative design preferences in RSPL and assess its feature performance, with the potential to enhance applications in residential planning and urban planning development. By analyzing design and feature performance choices, government authorities can identify the most promising urban planning areas. The following conclusions and reflections can be drawn from this experimental study on a case study of residential areas in China:

1.  The experimental framework of the "extraction-translation-machine learning-evaluation" proposed in this study addressed the deficiency of simultaneously considering all design elements of residential areas within the same methodological framework. This methodological framework integrated both machine and manual computations, as well as quantitative and qualitative evaluation techniques, to jointly determine research outcomes and comprehensively characterize the scientific nature of this study.

Furthermore, this experimental framework established a methodological paradigm for machine learning-assisted plan layout explorations.

2. Machine learning favors the generation of a balanced layout and showcases the innovative design potential of various elements in harmony with housing design components. When comparing the residential area before and after machine learning, it was observed that the generated plan exhibited less fluctuation in terms of building density, floor area ratio, and active land ratio compared to the original plan. Furthermore, the comparison of two design elements, square paving and green landscape space, reveals that machine learning aligns well with the building layout and offers innovative and diverse design perspectives. This, in turn, provides inspirational ideas for residential area layout design and promotes the enhancement of environmental quality within the residential area.

3. Machine learning exhibits a more pronounced generative preference for two design elements: other public facilities and spatial structures. When comparing the generated designs before and after machine learning, there was an increase in the number of design elements. RGB pixels were assigned to form large blocks of other public facilities and spatial structures that were connected and distributed in fragments. Furthermore, the machine-learned design element of other public facilities highlights the master-centered nature of the site. In the process of learning spatial structure, both monocentric and polycentric characteristics of residential spatial structures were generated, resulting in various forms of spatial structure design. Ultimately, this can aid planners in developing schemes that better align with residents' expectations. It also contributes to the discipline of urban planning by offering design ideas for the layout of urban infrastructure, public facilities, landscaped green spaces, and diverse spatial configurations.

**Author Contributions:** Conceptualization, P.S. and F.Y.; methodology, P.S. and F.Y.; software, P.S. and Q.H.; validation, P.S. and Q.H.; formal analysis, P.S.; investigation, P.S.; resources, P.S.; data curation, P.S.; writing—original draft preparation, P.S.; writing—review and editing, P.S., F.Y. and H.L.; visualization, P.S. and Q.H.; supervision, F.Y.; funding acquisition, F.Y. All authors have read and agreed to the published version of the manuscript.

**Funding:** This research was funded by the National Natural Science Foundation of China, grant number 42341207.

**Data Availability Statement:** Not applicable.

**Conflicts of Interest:** The authors declare no conflict of interest.

## Appendix A

**Table A1.** Residential schemes collection website source.

| | | |
|---|---|---|
| | https://www.om.cn/ | accessed on 9 April 2022 |
| Residential area scheme collection website source | https://www.doczhi.com/ | accessed on 16 April 2022 |
| | https://www.gstarcad.com/ | accessed on 23 April 2022 |
| | https://www.znzmo.com/ | accessed on 28 April 2022 |

**Table A2.** Five sets of experiment generated results were scored.

| Score Information | | Group A Score | Group B Score | Group C Score | Group D Score | Group E Score |
|---|---|---|---|---|---|---|
| Score Identity | Number | | | | | |
| Non-urban planning major students | 1 | 3.6 | 2.8 | 3.8 | 3.2 | 4.2 |
| | 2 | 1.5 | 2.7 | 3.1 | 3.4 | 3.9 |

<div align="center">**Table A2.** *Cont.*</div>

| Score Information | | Group A Score | Group B Score | Group C Score | Group D Score | Group E Score |
|---|---|---|---|---|---|---|
| Score Identity | Number | | | | | |
| Non-urban planning major students | 3 | 3.5 | 2.9 | 3.2 | 2.3 | 3.4 |
| | 4 | 2.5 | 2.3 | 2.7 | 3.0 | 3.2 |
| | 5 | 3.9 | 3.7 | 4.0 | 4.2 | 4.5 |
| | 6 | 2.6 | 2.6 | 3.3 | 3.6 | 3.8 |
| | 7 | 3.7 | 3.4 | 3.8 | 4.1 | 4.8 |
| | 8 | 1.9 | 2.3 | 3.4 | 3.7 | 4.4 |
| | 9 | 2.8 | 2.9 | 3.5 | 3.1 | 3.6 |
| | 10 | 3.8 | 4.2 | 4.1 | 4.3 | 4.7 |
| | 11 | 2.7 | 3.6 | 3.4 | 3.8 | 4.1 |
| | 12 | 1.6 | 2.4 | 3.0 | 3.3 | 3.7 |
| | 13 | 0.8 | 1.3 | 2.5 | 2.7 | 3.1 |
| | 14 | 2.1 | 2.5 | 2.8 | 3.1 | 3.4 |
| | 15 | 1.1 | 1.6 | 2.1 | 2.6 | 2.9 |
| Urban planning major students | 16 | 2.0 | 1.8 | 2.3 | 3.5 | 3.9 |
| | 17 | 2.3 | 3.0 | 3.8 | 3.2 | 4.2 |
| | 18 | 1.7 | 2.9 | 2.3 | 3.7 | 3.8 |
| | 19 | 2.6 | 2.1 | 3.7 | 3.1 | 3.9 |
| | 20 | 1.9 | 2.8 | 3.6 | 4.3 | 4.7 |
| | 21 | 2.3 | 2.4 | 3.3 | 3.7 | 3.9 |
| | 22 | 2.6 | 2.7 | 3.1 | 3.8 | 4.1 |
| | 23 | 1.2 | 2.1 | 2.7 | 3.3 | 3.7 |
| | 24 | 2.5 | 2.9 | 2.4 | 3.1 | 3.5 |
| | 25 | 2.8 | 3.6 | 3.4 | 4.1 | 4.3 |
| | 26 | 2.4 | 3.2 | 3.9 | 4.5 | 4.7 |
| | 27 | 1.4 | 2.2 | 2.7 | 3.0 | 3.3 |
| | 28 | 2.2 | 3.6 | 3.4 | 4.2 | 4.6 |
| | 29 | 1.8 | 2.5 | 3.3 | 2.2 | 3.6 |
| | 30 | 0.9 | 1.8 | 2.9 | 2.7 | 3.2 |
| average value | | 2.25 | 2.3 | 3.35 | 2.95 | 3.7 |

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
