# Peer review of "The Development of an Experimental Framework to Explore the Generative Design Preference of a Machine Learning-Assisted Residential Site Plan Layout"

_land, doi:10.3390/land12091776_

Round 1

Reviewer 1 Report

 The generative design process (RSPL) of residential site layout involved in this paper is a field of great research significance. The article uses the Pix2pix model to test several residential communities in China, attempting to construct an experimental framework of "extraction translation machine learning evaluation". The article adopts various mathematical calculation methods to determine the design elements and feature representations of machine learning design preferences in the field of "RSPL", and has achieved considerable research results. The contribution of this article is to address the preference issues of various design elements in residential areas within the same methodological framework, which can help planners develop solutions that better meet residents' expectations and elucidate the potential and beneficial directions of machine learning assisted "RSPL" applications.

 The article is logically coherent and reasonable, with clear research objectives and appropriate methods. But after reading, I have some minor doubts about some of the content of this article, which will be discussed in the following sections.

 1. The introduction section of the article is too long

 In the first part of the introduction, the article provides a large number of statements on the concepts involved, including the definition of RSPL, the significance of machine learning, and the definition of generating adversarial networks. However, after reading the entire article, I believe that the content in the introduction is biased towards a popular science introduction, and for a mature paper, the introduction section can be further refined.

 2. The reasons of using the pix2pix model as a research model and optimization direction

 In the introduction section, the comparison between the Pix2pix model and other similar models is relatively short. As the core model of this study, Pix2pix requires a more detailed description of its advantages to demonstrate the necessity of using it for subsequent analysis. At the same time, listing the advantages and disadvantages of each model can also serve as a possible direction for model optimization. In addition, the article has conducted sufficient research on RSPL based on the existing Pix2pix model and achieved considerable results. In the end of the third part, it is mentioned that the Pix2pix model may have performance optimization in the future, so it can make certain prospects for the direction of model optimization and the development of RSPL.

 3.Display typical residential plans

 The research objects of the article are nearly 300 Chinese residential plans, and there should be some particular selection criteria for the selection of research objects, such as area range, residential type, etc. At the same time, several typical untreated residential floor plans should also be displayed to assist in reading and understanding the research process.

 Personally, I believe that the main existing issues in the article are basically the same, and there is still room for further optimization of the research steps. In addition, there are a series of minor issues, as listed below:

 I sincerely hope my opinions can be adopted and revised. Thanks for your reading.

 1. Line 247- Missing space after 'where'.

 2. Line 251 - Why is the SSIM number still b when the b number has already been used in the previous text.

 3. Line 463~469 - Table and text misalignment.

Author Response

On behalf of all the contributing authors, I am expressing my sincere gratitude to the reviewers for their valuable suggestions, which helped improve our manuscript. Based on your comments, we have revised the manuscript. A point-by-point response to the reviewers is listed in this letter. In this revised version, the changes to our manuscript are highlighted in red text in the document but do not affect the content or framework of the paper. We sincerely appreciate your work and hope that the revisions will be recognized. Thank you again for your suggestions and comments.

Reviewer 2 Report

The authors attempted a novel approach by employing the pix2pix model to aid RSPL. However, there are pivotal inquiries that I wish the author to address and clarify.

1.     The quality of Tables and Figures needs significant improvement. For instance, Tables 4 and 6 fail to effectively convey the specific changes in the samples after applying the proposed model. Improvements are needed in the graphical representation of images. On Page 16, the table lacks appropriate numbering, and its purpose remains ambiguous. Although an abundance of information is presented, its direct significance is not evident. Figures 7-9 also lack clarity in terms of their contents and the intended information they aim to convey.

2.     During the actual design phase, external limiting factors necessitate consideration, such as land use regulations, budgetary constraints, as well as local cultural and economic dynamics. It is insufficient to merely employ machine learning to randomly generate designs, as this approach overly idealizes the design process and lacks adequate human involvement. Consequently, the investigation of "the design preference of machine learning" raises concerns regarding its necessity, particularly from my perspective.

3.     Despite the extensive presentation of results throughout the article, I remain considerably perplexed. Is there any real practical applicability? Given the potential for a different model to yield distinct designs and preferences, it raises questions about which approach/model can truly be feasibly adopted. It is my hope that the authors will contemplate the underlying rationale for incorporating Machine Learning within this domain, as well as the added value that ML brings, rather than just employing it without any consideration.

The English is readable but still needs improvement.

Author Response

(The authors gave the same response as above.)

Reviewer 3 Report

This is a very interesting paper, focusing on generative design based on machine learning by developing an experimental framework to explore the generative design preference of machine learning -assisted residential site plan layout. The approach and methodology presented would be very interesting for readers working in the associated field and the conclusions derived from the study are also offering some very useful insights. I do not have any further comments or suggestions for improvement of this paper. Excellent contribution to knowledge in the related field.

Author Response

We are very grateful to the reviewers for recognizing and appreciating our research. We will continue our efforts in the future.

Round 2

Reviewer 2 Report

Thanks for the detailed responses from the authors. I suggest that the paper can be published.

Minor comment is as below:

1.     Reference 13 and 48, please provide an English translation. Please check the reference format as well as I spotted inconsistency.

English can be improved to be more precise.
